# Adaptively Phased Algorithm for Linear Contextual Bandits

## Abstract

We propose a novel algorithm for the linear contextual bandit problem when the set of arms is finite. Recently the minimax expected regret for this problem is shown to be $\Omega(\sqrt{dT\log T \log K})$ with $T$ rounds, $d$-dimensional contexts, and $K \leq 2^{d/2}$ arms per time. Previous works on phased algorithms attain this lower bound in the worst case up to logarithmic factors (Auer, 2002; Chu et al., 2011) or iterated logarithmic factors (Li et al., 2019), but require a priori knowledge of the time horizon $T$ to construct the phases, which limits their use in practice. In this paper we propose a novel phased algorithm that does not require a priori knowledge of $T$, but constructs the phases in an adaptive way. We show that the proposed algorithm guarantees a regret upper bound of order $O(d^\alpha \sqrt{T\log T (\log K + \log T)})$ where $\frac{1}{2} \leq \alpha \leq 1$. The proposed algorithm can be viewed as a generalization of Rarely Switching OFUL (Abbasi-Yadkori et al., 2011) by capitalizing on a tight confidence bound for the parameter in each phase obtained through independent rewards in the same phase.

## 1 Introduction

We consider a sequential decision problem where the learning agent is repeatedly faced with a set of available actions, chooses an action from this set, and receives a random reward. We assume that the expected value of the reward is an unknown linear function of the context information of the chosen action. This is a linear contextual bandit problem, where taking actions is characterized as pulling the arms of a bandit slot machine. At each time step, the learner has access to the context vector of each arm. The goal of the learner is to minimize the cumulative gap between the rewards of the optimal arm and the chosen arm, namely regret. Due to uncertainty about the compensation mechanism of the reward, the learner should balance the trade-off between exploitation, pulling the seemingly best arm based on the function learned so far, and exploration, pulling other arms that would help learn the unknown function more accurately.

Auer (2002), Li et al. (2010), and Abbasi-Yadkori et al. (2011) proposed the LinRel, LinUCB, and OFUL algorithms, respectively, for the linear bandit problem. The underlying principle of these algorithms is optimism-in-the-face-of-uncertainty, where the arms are pulled to maximize the optimistic estimate of the expected reward. Based on the same principle, Auer (2002), Chu et al. (2011) and Li et al. (2019) presented phased algorithms, which are different from the aforementioned methods in two aspects: the number of updates for the estimate is smaller, and the order of regret upper bound in terms of the dimension of contexts is smaller. In this paper, we focus on the phased algorithms.

Most algorithms update the estimates every round and pull the arm based on the most up-to-date estimate. This does not raise an issue when the computation of the estimates is not costly, however, could become an issue when computation is heavy. Recently, contextual bandit algorithm is proposed for deep neural networks (Zhou et al., 2020; Zhang et al., 2021) requiring computation of a deep model every round. Similarly, for bandit applications attempting to enhance deep learning such as hyperparameter search (Li et al., 2017) and active learning (Ganti & Gray, 2013), the rewards are error rates of the fitted model. In some of these applications, updating the estimates every round may not be practically feasible. There have been a number of phased algorithms, SupLinRel (Auer, 2002), SupLinUCB (Chu et al., 2011), and Variable-Confidence-Level SupLinUCB (Li et al., 2019). The central idea of the phased algorithms is to assign the decision steps

into separate $\lceil \log T \rceil$ phases. In phased algorithms, the estimates are not updated within the same phase until the uncertainty estimates decrease below a predefined level. Less frequent updates of phased algorithms offer substantial advantages when the computation of the estimates is costly.

As for the order of regret upper bound, Dani et al. (2008) showed that when contexts are $d$-dimensional, there exists a distribution of contexts such that any algorithm has regret at least $\Omega(d\sqrt{T})$, where $T$ is the time horizon. Under additional assumption that the number of arms $K$ in each step is finite, Chu et al. (2011) showed a lower bound of $\Omega(\sqrt{dT})$ and proposed an algorithm which matches this bound up to poly-logarithmic factors in the worst case. Recently, Li et al. (2019) refined this lower bound to $\Omega(\sqrt{dT \log T \log K})$, showing that some logarithmic terms are unavoidable for any algorithm.

On the other hand the OFUL (Abbasi-Yadkori et al., 2011) and the LinUCB (Li et al., 2010) algorithms have a regret upper bound of order $O(d\sqrt{T}\log T)$ regardless of the number of arms. When the number of arms is large as $K \geq 2^{d/2}$, the regrets of OFUL and LinUCB match the best lower bound up to logarithmic factors. However when $K$ is small ($K \ll 2^{d/2}$), the regrets are larger by a factor of $O(\sqrt{d})$ compared to the best possible bound.

The extra $O(\sqrt{d})$ term appears in the estimation error bound of the linear regression parameter due to dependence of chosen contexts on the rewards. Since LinUCB and OFUL update the arm choice rule in each round based on the rewards observed up to that round, the chosen contexts are correlated with the rewards. To some extent, such correlation is necessary because we cannot minimize the regret without any adaptation to the past observations unless we completely know the parameter values. However, we can carefully control the amount of correlation to make use of the tighter confidence result. Aforementioned phased algorithms have addressed this issue by handling computation separately for each phase. In these algorithms the arms in the same phase are chosen without making use of the rewards in the same phase. Consequently in each phase, due to independence, a tight confidence bound could be constructed for the parameter without extra factor $O(\sqrt{d})$.

Despite the strong theoretical guarantee, the aforementioned phased algorithms, SupLinRel, SupLinUCB, and VCL SupLinUCB, are not practical since the implementation requires a priori knowledge of the time horizon $T$ to determine the number of phases. A loose upper bound of $T$ could result in a larger number of phases than it is needed, reducing the number of samples in each phase and increasing the regret. Another difficulty is that the phases switch around over the time round. The doubling trick (Auer et al., 1995) can be used, but is wasteful. Even under the knowledge of $T$, the algorithms are mostly outperformed by LinUCB (Valko et al., 2014).

In this paper, we propose a novel phased algorithm where changes in phases are monotone in time. The proposed method does not require a priori knowledge of time horizon $T$ since the switching time of the phase is determined in an adaptive way and the number of phases is determined accordingly. At a given phase, the arms are pulled based on the estimate from the previous phase. Hence, the chosen contexts are not correlated with the rewards of the current phase. We do not switch phases until the upper bound of the regret in that phase exceeds a predefined constant. The proposed algorithm achieves a regret upper bound of $O(d^{\alpha}\sqrt{T\log T}(\log K + \log T))$, where $\alpha \in [\frac{1}{2}, 1]$ depends on the number of phases.

## 1.1 Related works

A number of authors have studied phased algorithms that save computational cost. When Abbasi-Yadkori et al. (2011) proposed OFUL in their paper, they also proposed a phased algorithm called Rarely Switching OFUL to recompute the estimates only $O(\log T)$ times. However, their regret upper bound is $O(d\sqrt{T}\log T)$. SupLinRel (Auer, 2002) and SupLinUCB (Chu et al., 2011) both achieve a regret upper bound of $O(\sqrt{dT}\log^{3/2}(KT\log T))$, removing the extra $O(\sqrt{d})$ factor when $K \ll 2^{d/2}$. The recently proposed Variable-Confidence-Level (VCL) SupLinUCB (Li et al., 2019) refines the SupLinUCB and achieves a tighter bound, $O(\sqrt{dT \log T \log K}(\log\log T)^{\gamma})$, with $\gamma > 0$. The difference between Rarely Switching OFUL and others is that the estimates in Rarely Switching OFUL are based on the data from the beginning to the previous phase, while SupLinRel, SupLinUCB and VCL SupLinUCB estimate the parameter in each phase separately.

Valko et al. (2014) and Lattimore & Szepesvári (2020) also proposed phase-wise algorithms with $O(\sqrt{dT}\log K\log T)$ regret guarantees. Their methods however are restricted to the cases where the set of arms is fixed over time. Both works use the phase-wise arm elimination idea of Auer & Ortner (2010), eliminating subotpimal arms at the end of each phase. Due to elimination, the maximum possible regret decreases after each phase. In each phase, the algorithm either pulls the most uncertain arms (Valko et al., 2014) or pulls each arm according to an optimal design strategy (Lattimore & Szepesvári, 2020), without any dependence on the rewards of the current phase. In this paper, we allow the arm set to change arbitrarily over time.

### 1.2 Contributions

The main contributions of the paper are as follows.

- We propose a novel phased algorithm for the linear contextual bandit problem where the estimates are updated only $O(\log T)$ times and a tight confidence bound for the linear parameter is used.

- The proposed algorithm does not require prior knowledge of $T$. The changes in phases are monotone in time, and the number of phases is determined in an adaptive mechanism.

- We prove that the high-probability regret upper bound of the proposed algorithm ranges between $O(d\sqrt{T\log T}(\log K + \log T))$ and $O(\sqrt{dT\log T}(\log K + \log T))$, depending on the number of phases.

## 2 Problem formulation

At each time $t$, the learner is faced with $K$ alternative arms. The $i$-th arm ($i = 1, \cdots, K$) yields a random reward $r_i(t)$ with unknown mean. Prior to the choice at time $t$, the learner has access to a finite-dimensional context vector $b_i(t) \in \mathbb{R}^d$ associated with each arm $i$. Then the learner pulls one arm $a(t)$ and observes the corresponding reward $r_{a(t)}(t)$. We also make the following assumptions, from A1 to A4.

**A1. Linear reward.** *For an unknown vector $\mu \in \mathbb{R}^d$,*

$$\mathbb{E}[r_i(t)|b_i(t)] = b_i(t)^T\mu.$$

**A2. Bounded norms.** *Without loss of generality, $||b_i(t)||_2 \leq 1$, $||\mu||_2 \leq 1$.*

**A3. Sub-Gaussian error.** *The error $\eta_i(t) := r_i(t) - b_i(t)^T\mu$ is $R$-sub-Gaussian for some $R > 0$, i.e., for every $\epsilon \in \mathbb{R}$,*

$$\mathbb{E}[\exp(\epsilon\eta_i(t))] \leq \exp(\epsilon^2 R^2/2).$$

**A4. Oblivious adversary.** *The sequence of contexts is chosen by an oblivious adversary. An oblivious adversary may know the algorithm's code, but does not have access to the randomized results of the algorithm.*

Assumption A4 is used in Auer (2002), Chu et al. (2011), and Li et al. (2019) which consider the same problem as ours. Under assumption A1, the optimal arm at time $t$ is $a^*(t) := \text{argmax}_i\{b_i(t)^T\mu\}$. We define the $regret(t)$ as the difference between the expected reward of the optimal arm and the expected reward of the arm chosen by the learner at time $t$, i.e.,

$$regret(t) = b_{a^*(t)}(t)^T\mu - b_{a(t)}(t)^T\mu.$$

Then, the goal of the learner is to minimize the sum of regrets over $T$ steps, $R(T) := \sum_{t=1}^{T} regret(t)$.

## 3 Proposed method

Our strategy is to adaptively combine methods for (B1) and (B2) described below, and the derivation of the adapting conditions is the key of this Section. The phased algorithms use (B1) which yields a tighter

confidence bound but they require the knowledge of $T$, and the Rarely Switching OFUL invokes (B2) adaptively but with a wider bound. The two bounds play a crucial role in deriving the phase-switching conditions for our method. In Section 3.1, we first review the difference in the prediction error bounds of the estimate for the expected reward when, within the phase,

(B1) arms are chosen independently of the rewards,

(B2) arms are chosen adaptively based on the rewards observed so far.

Then in Section 3.2, we present a new class of phase-wise Upper Confidence Bound (UCB) algorithm and discuss on the phase-switching conditions to bound the regret. Finally in Section 3.3, we propose the adaptively phased Upper Confidence Bound (AP-UCB) algorithm.

### 3.1 Parameter estimation

Let $\mathcal{S}$ be the set of context-reward pairs of the chosen arms at time points in the set $\mathcal{T} \subset \mathbb{N}$, i.e., $\mathcal{S} = \{(b_{a(\tau)}(\tau), r_{a(\tau)}(\tau)), \tau \in \mathcal{T}\}$, where $\mathbb{N}$ is the set of natural numbers. We use the following Ridge estimator with some constant $\lambda > 0$ to estimate the linear parameter $\mu$.

$$\hat{\mu} = \Big(\lambda I_d + \sum_{(b_\tau, r_\tau) \in \mathcal{S}} b_\tau b_\tau^T\Big)^{-1} \sum_{(b_\tau, r_\tau) \in \mathcal{S}} b_\tau r_\tau.$$

Chu et al. (2011) and Abbasi-Yadkori et al. (2011) analyzed the upper bound of the prediction error $|b_i(t)^T(\hat{\mu} - \mu)|$ for case (B1) and (B2), respectively.

**Lemma 3.1. (Lemma 1 of Chu et al., 2011)** *Suppose that the samples in $\mathcal{S}$ are such that for fixed $b_{a(\tau)}(\tau)$ with $\tau \in \mathcal{T}$, the rewards $r_{a(\tau)}(\tau)$'s are independent random variables with means $\mathbb{E}[r_{a(\tau)}(\tau)] = b_{a(\tau)}(\tau)^T \mu$. Then for a fixed $t$ and for all $1 \le i \le K$, we have with probability at least $1 - \frac{\delta}{t^2}$,*

$$|b_i(t)^T(\hat{\mu} - \mu)| \le \Big(2R\sqrt{\log\big(\frac{2Kt}{\delta}\big)} + \sqrt{\lambda}\Big)s_{t,i},$$

*where $s_{t,i} = \sqrt{b_i(t)^T B^{-1} b_i(t)}$ and $B = \lambda I_d + \sum_{(b_\tau, r_\tau) \in \mathcal{S}} b_\tau b_\tau^T$.*

**Lemma 3.2. (Theorem 2 of Abbasi-Yadkori et al., 2011)** *Define $\mathcal{H}_{t-1}$ as the history until time $t - 1$, i.e., $\mathcal{H}_{t-1} = \{a(\tau), r_{a(\tau)}(\tau), \{b_i(\tau)\}_{i=1}^K, \tau = 1, \cdots, t-1\}$, and the filtration $\mathcal{F}_{t-1}$ as the union of $\mathcal{H}_{t-1}$, the contexts at time $t$, and the action at time $t$, i.e., $\mathcal{F}_{t-1} = \{\mathcal{H}_{t-1}, \{b_i(t)\}_{i=1}^K, a(t)\}$ for $t = 1, \cdots, T$. Suppose that the samples in $\mathcal{S}$ are such that for each $\tau \in \mathcal{T}$, $\mathbb{E}[r_{a(\tau)}(\tau)|\mathcal{F}_{\tau-1}] = b_{a(\tau)}(\tau)^T \mu$ and $\eta_{a(\tau)}(\tau)$ is conditionally $R$-sub-Gaussian given $\mathcal{F}_{\tau-1}$, i.e.,*

$$\mathbb{E}[\exp(\epsilon \eta_{a(\tau)}(\tau))|\mathcal{F}_{\tau-1}] \le \exp(\epsilon^2 R^2/2).$$

*Then for a fixed $t$ and for all $1 \le i \le K$, we have with probability at least $1 - \frac{\delta}{t^2}$,*

$$|b_i(t)^T(\hat{\mu} - \mu)| \le \Big(R\sqrt{3d\log\big(\frac{t}{\delta}\big)} + \sqrt{\lambda}\Big)s_{t,i}.$$

The bound in Lemma 3.2 does not depend on $K$, but has an extra $O(\sqrt{d})$ factor compared to Lemma 3.1. The key point in Lemma 3.1 is that the error $\hat{\mu} - \mu \approx B^{-1} \sum b_\tau \eta_{a(\tau)}(\tau)$ can be expressed as the sum of independent, mean zero variables. This is because $b_\tau$'s and $\eta_{a(\tau)}(\tau)$'s are independent so $b_\tau$'s and $B$ can be considered as fixed variables. Hence, we can directly apply the Chernoff inequality and obtain a tight bound. On the other hand, in Lemma 3.2, the error is not the sum of independent variables due to the correlation between the context variables inside $B$ and $\eta_{a(\tau)}(\tau)$'s. Hence, we should invoke the Cauchy-Schwarz inequality which gives two terms, one corresponding to $s_{t,i}$ and the other including the normalized sum of $\eta_{a(\tau)}(\tau)'s$ which can be bounded by the self-normalized theorem. Since each term contributes a factor of $\sqrt{d}$, the combined bound has an order of $d$.

### 3.2 Phase-switching condition

To make use of the tight confidence result in Lemma 3.1, we propose a phase-wise algorithm which updates the regression parameter only at the end of each phase. Algorithm 1 shows an example of such phase-wise linear bandit algorithm. The arm choice of the $m$-th phase depends on the estimate $\hat{\mu}_{m-1}$ and matrix $B_{m-1}$ constructed in the preceding phase. Hereby, the estimate of each phase has a small prediction error. There are two differences comparing with LinUCB: first it updates the estimate of $\mu$ and the matrix $B$ infrequently, second, the estimates are based on the data from the previous phase as marked by lines 8–11 in the algorithm. Comparing with Rarely Switching OFUL, the difference lies in line 11, where the set $\mathcal{S}$ stores data only from the current phase. For now, we do not specify the phase-switching condition but simply denote the end point of the $m$-th phase as $t_m$.

---

**Algorithm 1** phase-wise UCB

---

1: Input: $\alpha$, $\lambda$
2: Set: $\mathcal{S} = \{\}$, $m = 1$, $\hat{\mu}_0 = 0_d$, $B_0 = \lambda I_d$
3: **for** $t = 1, \cdots, T$ **do**
4:    Pull arm $a(t) = \text{argmax}_{1 \leq i \leq K}\{b_i(t)^T \hat{\mu}_{m-1} + \alpha s_{t,i}\}$ where $s_{t,i} = \sqrt{b_i(t)^T B_{m-1}^{-1} b_i(t)}$
5:    Observe reward $r_{a(t)}(t)$
6:    $\mathcal{S} \leftarrow \mathcal{S} \cup \{(b_{a(t)}(t), r_{a(t)}(t))\}$
7:    **if** $t = t_m$ **then**
8:      $B_m \leftarrow \lambda I_d + \sum_{(b_\tau, r_\tau) \in \mathcal{S}} b_\tau b_\tau^T$
9:      $\hat{\mu}_m \leftarrow B_m^{-1} \sum_{(b_\tau, r_\tau) \in \mathcal{S}} b_\tau r_\tau$
10:      $m \leftarrow m + 1$
11:      $\mathcal{S} \leftarrow \{\}$
12:    **end if**
13: **end for**

---

We derive an upper bound of the regret of Algorithm 1 and deduce the phase-switching condition that minimizes this upper bound. Let $\alpha = 2R\sqrt{\log\left(\frac{2KT}{\delta}\right)} + \sqrt{\lambda}$. Consider time $t$ in the $m$-th phase, i.e., $t_{m-1} < t \leq t_m$ with $t_0 = 0$. We have with probability at least $1 - \frac{\delta}{t^2}$,

$$
\begin{aligned}
b_{a^*(t)}(t)^T \mu &\leq b_{a^*(t)}(t)^T \hat{\mu}_{m-1} + \alpha s_{t,a^*(t)} \\
&\leq b_{a(t)}(t)^T \hat{\mu}_{m-1} + \alpha s_{t,a(t)} \\
&\leq b_{a(t)}(t)^T \mu + \alpha s_{t,a(t)} + \alpha s_{t,a(t)},
\end{aligned}
\tag{1}
$$

where the first and third inequalities are due to Lemma 3.1 and the second inequality is due to the arm selection mechanism. Therefore,

$$
regret(t) = b_{a^*(t)}(t)^T \mu - b_{a(t)}(t)^T \mu \leq 2\alpha s_{t,a(t)}.
\tag{2}
$$

Applying the union bound to all time points, we have with probability at least $1 - \delta$,

$$
R(T) \leq 2\alpha \sum_{t=1}^{T} s_{t,a(t)} = 2\left(2R\sqrt{\log(\frac{2KT}{\delta})} + \sqrt{\lambda}\right) \sum_{t=1}^{T} s_{t,a(t)}.
$$

In LinUCB, the matrix $B$ at time $t$ is the Gram matrix of all the chosen contexts up to time $t-1$, $B(t) = \lambda I_d + \sum_{\tau=1}^{t-1} b_{a(\tau)}(\tau) b_{a(\tau)}(\tau)^T$, and the sum $\sum_{t=1}^{T} s_{t,a(t)}$ can be shown to be less than $O(\sqrt{dT\log T})$ by the elliptical potential lemma of Abbasi-Yadkori et al. (2011). However in the phase-wise algorithm, we always have $B_{m-1} \preccurlyeq B(t)$ for any $t_{m-1} < t \leq t_m$. Therefore, the elliptical potential lemma cannot apply. We have

instead,

$$\sum_{t=1}^{T} s_{t,a(t)} = \sum_{m=1}^{M} \sum_{t=t_{m-1}+1}^{t_m} \sqrt{b_{a(t)}(t)^T B_{m-1}^{-1} b_{a(t)}(t)} \leq \sqrt{T \sum_{m=1}^{M} \sum_{t=t_{m-1}+1}^{t_m} b_{a(t)}(t)^T B_{m-1}^{-1} b_{a(t)}(t)}$$

$$= \sqrt{T \sum_{m=1}^{M} trace\Big(B_{m-1}^{-1} \sum_{t=t_{m-1}+1}^{t_m} b_{a(t)}(t) b_{a(t)}(t)^T\Big)} \leq \sqrt{T \sum_{m=1}^{M} trace\Big(B_{m-1}^{-1} B_m\Big)}, \qquad (3)$$

where $M$ denotes the total number of phases and the first inequality is due to Jensen's inequality.

The bound (3) motivates a phase-switching condition. Suppose we switch from the $m$-th phase to the $(m+1)$-th phase as soon as the trace of $B_{m-1}^{-1} B_m$ exceeds $Ad$ for some predefined constant $A > 0$. Then we can bound $\sum s_{t,a(t)} \leq \sqrt{MAdT}$, which has the same order as the bound of the elliptical potential lemma. The problem is that we do not have control over the number of phases, $M$. A sufficient condition for bounding $M$ is that the determinant of $B_m$ is larger than the determinant of $CB_{m-1}$ for some predefined constant $C > 1$. If $det(B_m) \geq det(CB_{m-1})$ for all $1 \leq m \leq M$, we have,

$$det(B_M) \geq det(CB_{M-1}) \geq det(C^2 B_{M-2}) \geq \cdots \geq det(C^M B_0).$$

Thus, $det(C^M \lambda I_d) \leq det(B_M)$, with $det(C^M \lambda I_d) = (\lambda C^M)^d$ and

$$det(B_M) \leq det\Big(\lambda I_d + \sum_{t=1}^{T} b_{a(t)}(t) b_{a(t)}(t)^T\Big) \leq \Big(\frac{\lambda d + \sum_{t=1}^{T} b_{a(t)}(t)^T b_{a(t)}(t)}{d}\Big)^d \leq \big(\lambda + \frac{T}{d}\big)^d,$$

where the second inequality is due to the determinant-trace inequality and the third inequality due to Assumption A2. Therefore, $M$ can be kept small as $M \leq \log_C\big(1 + T/d\lambda\big)$.

When the two conditions (C1) $trace(B_{m-1}^{-1} B_m) \leq Ad$ and (C2) $det(B_m) \geq det(CB_{m-1})$ are satisfied for every $m$, the regret of the proposed algorithm achieves a tight regret upper bound, $R(T) \leq O(\sqrt{dT \log T}(\log K + \log T))$. However, imposing (C1) does not always guarantee (C2). In the next section, we suggest a remedy when (C2) does not hold.

### 3.3 Algorithm

To check conditions (C1) and (C2) at phase $m$, we do not need to observe the rewards of the chosen arms. We can *choose* the arms based on $\hat{\mu}_{m-1}$ given in line 9 of Algorithm 1, and compute $\sum_{t=t_{m-1}+1}^{t_m} b_{a(t)}(t) b_{a(t)}(t)^T$ until condition (C1) is violated. Then at that round, if condition (C2) holds, we can *pull* the chosen arms. We call this an independent phase, because it uses information only from the preceding phase. If condition (C2) does not hold, we can go back to the beginning of the current phase, and pull the arms based on $\tilde{\mu}_{m-1}$ given in line 28 in Algorithm 2, which uses samples in all cumulative phases. We call this phase an adaptive phase. For the adaptive phase, we use the phase switching condition of Rarely Switching OFUL (Abbasi-Yadkori et al., 2011), which is given in line 21 of Algorithm 2 and is different from the switching condition for the independent phase.

In every phase, we start with an independent phase using the estimate from the most recent independent phase. Algorithm 2 presents the proposed AP-UCB algorithm. We remark that when condition (C2) is never met, AP-UCB is identical to the Rarely Switching OFUL algorithm which guarantees a regret less than $O(d\sqrt{T}\log T)$. In contrast when condition (C2) is always satisfied, the regret is less than $O(\sqrt{dT \log T}\log(KT))$ as we have seen in Section 3.2.

## 4 Regret analysis

The following theorem derives a high-probability upper bound of the regret incurred by the proposed algorithm in terms of the number of independent phases, $M_1$, and the number of adaptive phases, $M_2$.

---

**Algorithm 2** AP-UCB

---

1: Input: $\alpha, \beta, \lambda, A > 0, C > 1, E > 1$
2: Set: $\hat{\mu}_0 = \tilde{\mu}_0 = 0_d$, $B_0 = B'_0 = \lambda I_d$, $t = t_0 = 0$, $\bar{m} = 1$.
3: **for** $m = 1, \cdots$ **do**
4:     $B \leftarrow \lambda I_d$
5:     **while** $trace(B_{\bar{m}-1}^{-1}B) \leq Ad$ **do**
6:         $t \leftarrow t + 1$
7:         Choose arm $a(t) = \text{argmax}_i\{b_i(t)^T\hat{\mu}_{\bar{m}-1} + \alpha s_{t,i}\}$ where $s_{t,i} = \sqrt{b_i(t)^T B_{\bar{m}-1}^{-1} b_i(t)}$
8:         $B \leftarrow B + b_{a(t)}(t)b_{a(t)}(t)^T$
9:     **end while**
10:    **if** $det(B) \geq det(CB_{\bar{m}-1})$ **then**
11:        $t_m \leftarrow t$
12:        Pull arms $\{a(\tau)\}_{\tau=t_{m-1}+1}^{t_m}$ and observe $\{r_{a(\tau)}(\tau)\}_{\tau=t_{m-1}+1}^{t_m}$.
13:        $B_{\bar{m}} \leftarrow B$
14:        $\hat{\mu}_{\bar{m}} \leftarrow B_{\bar{m}}^{-1}\sum_{\tau=t_{m-1}+1}^{t_m} b_{a(\tau)}(\tau)r_{a(\tau)}(\tau)$
15:        $B'_m \leftarrow B'_{m-1} + \sum_{\tau=t_{m-1}+1}^{t_m} b_{a(\tau)}(\tau)b_{a(\tau)}(\tau)^T$
16:        $\tilde{\mu}_m \leftarrow B_m'^{-1}\{B'_{m-1}\tilde{\mu}_{m-1} + \sum_{\tau=t_{m-1}+1}^{t_m} b_{a(\tau)}(\tau)r_{a(\tau)}(\tau)\}$
17:        $\bar{m} \leftarrow \bar{m} + 1$
18:    **else**
19:        $t \leftarrow t_{m-1}$
20:        $B'_m \leftarrow B'_{m-1}$
21:        **while** $det(B'_m) \leq Edet(B'_{m-1})$ **do**
22:            $t \leftarrow t + 1$
23:            Choose arm $\tilde{a}(t) = \text{argmax}_i\{b_i(t)^T\tilde{\mu}_{m-1} + \beta s'_{t,i}\}$ where $s'_{t,i} = \sqrt{b_i(t)^T B_{m-1}'^{-1} b_i(t)}$
24:            $B'_m \leftarrow B'_m + b_{\tilde{a}(t)}(t)b_{\tilde{a}(t)}(t)^T$
25:        **end while**
26:        $t_m \leftarrow t$
27:        Pull arms $\{\tilde{a}(\tau)\}_{\tau=t_{m-1}+1}^{t_m}$ and observe $\{r_{\tilde{a}(\tau)}(\tau)\}_{\tau=t_{m-1}+1}^{t_m}$.
28:        $\tilde{\mu}_m \leftarrow B_m'^{-1}\{B'_{m-1}\tilde{\mu}_{m-1} + \sum_{\tau=t_{m-1}+1}^{t_m} b_{\tilde{a}(\tau)}(\tau)r_{\tilde{a}(\tau)}(\tau)\}$
29:    **end if**
30: **end for**

---

**Theorem 4.1. Regret of AP-UCB**. *Suppose assumptions A1, A2, A3, and A4 hold. If we set* $\alpha = 2R\sqrt{\log(\frac{2KT}{\delta})} + \sqrt{\lambda}$ *and* $\beta = R\sqrt{3d\log(\frac{T}{\delta})} + \sqrt{\lambda}$ *for some* $\delta \in (0, 1)$, *we have with probability at least* $1 - 2\delta$,

$$R(T) \leq \sqrt{16dT\left\{2M_1AR^2\log\left(\frac{2KT}{\delta}\right) + 3M_2ER^2\log E\log\left(\frac{T}{\delta}\right)\right\}} + 16T\lambda$$
$$= O\left(\sqrt{dT(\log K + \log T)(M_1 + M_2)}\right).$$

*A sketch of proof.* We first have,

$$R(T) = \sum_{m=1}^{M}\sum_{t=t_{m-1}+1}^{t_m} regret(t) \leq \sqrt{T\sum_{m=1}^{M}\sum_{t=t_{m-1}+1}^{t_m} regret(t)^2},$$

where the inequality follows from Jensen's inequality. When $m$ is the $\bar{m}$-th independent phase, arms are chosen based on $\hat{\mu}_{\bar{m}-1}$ and $B_{\bar{m}-1}$ from the $(\bar{m} - 1)$-th independent phase. Using Lemma 3.1 and similar

arguments as in (1) and (2), we can show that with probability at least $1 - \delta$,

$$\sum_{t=t_{m-1}+1}^{t=t_m} regret(t)^2 \leq \sum_{t=t_{m-1}+1}^{t=t_m} 4\alpha^2 b_{a(t)}(t)^T B_{\bar{m}-1}^{-1} b_{a(t)}(t)$$

$$= 4\alpha^2 trace\Big(B_{\bar{m}-1}^{-1} \sum_{t=t_{m-1}+1}^{t=t_m} b_{a(t)}(t) b_{a(t)}(t)^T\Big) \leq 4\alpha^2 A d$$

for every independent phase, where the last inequality follows from condition (C1). On the other hand, when $m$ is an adpative phase, arms are chosen based on $\tilde{\mu}_{m-1}$ and $B'_{m-1}$ constructed from all cumulative samples up to the $(m-1)$-th phase. Using Lemma 3.2 instead of Lemma 3.1, we have with probability at least $1 - \delta$,

$$\sum_{t=t_{m-1}+1}^{t=t_m} regret(t)^2 \leq \sum_{t=t_{m-1}+1}^{t=t_m} 4\beta^2 b_{\tilde{a}(t)}(t)^T B_{m-1}^{'-1} b_{\tilde{a}(t)}(t) \leq 8\beta^2 E \log E$$

for every adaptive phase, where the last inequality is due to phase-switching condition for the adaptive phase. Therefore, with probability at least $1 - 2\delta$,

$$R(T) \leq \sqrt{T\{M_1 4\alpha^2 A d + M_2 8\beta^2 E \log E\}}.$$

Plugging in the definition of $\alpha$ and $\beta$ gives the theorem. □

Lemma 4.2 shows the upper bounds of $M_1$ and $M_2$. While $M_1$ is at most $O(\log T)$, the biggest possible value of $M_2$ scales with $O(d \log T)$, introducing an extra $O(\sqrt{d})$ factor to the regret bound. However, $M_2$ reaches the upper bound when the AP-UCB consists of only adaptive phases without independent phases. The lemma implies that we can avoid an extra $O(\sqrt{d})$ factor by keeping $M_2$ small and $M_1$ as large as possible.

**Lemma 4.2.** *In the AP-UCB algorithm, we have*

$$M_1 \leq \log_C\big(1 + T/d\lambda\big), \quad M_2 \leq d \log_E\big(1 + T/d\lambda\big).$$

Detailed proofs of Theorem 4.1 and Lemma 4.2 are presented in the Supplementary Material.

## 5 Experiments

We conduct simulation studies to compare the performance of the proposed algorithm with LinUCB and Rarely Switching (RS) UCB. We construct a similar environment to the design of Chu et al. (2011), where the distribution of the contexts and rewards is such that the regret is at least $\Omega(\sqrt{dT})$ for any algorithm. We set $K = 2$ and $d = 11$. Detailed structures are presented in the Supplementary material.

LinUCB and RS-UCB require an input parameter controlling the degree of exploration, which has the same theoretical order as $\beta$ of AP-UCB. RS-UCB and AP-UCB also share the parameter $E$ in the phase-switching condition for the adaptive phases. AP-UCB has additional hyperparameters, $\alpha, A$, and $C$. We fix $A = 1.5$, $C = 1.2$, and $E = 5$. We consider some candidate parameters for $\alpha$ and $\beta$ and report the results of the values that incur minimum median regret over 30 experiments.

Figure 1 shows the cumulative regret $R(t)$ according to time $t$. The proposed AP-UCB has the minimum median regret. LinUCB and RS-UCB have similar performance, which is in accordance with the theory. AP-UCB has 2 long independent phases, followed by 12 adaptive phases. In contrast, RS-UCB has 20 adaptive phases in total. Long independent phases may have led to a better performance of AP-UCB by gathering diverse context variables.

## 6 Concluding remarks

In this paper, we propose an adaptively phased algorithm for the linear contextual bandit problem with finitely many arms. The algorithm does not require a priori knowledge of the time horizon and saves

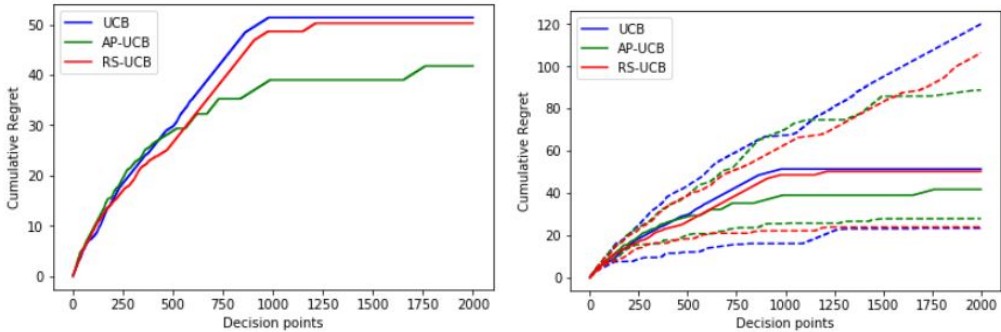

Figure 1: Median (solid), first and third quartiles (dashed) of the cumulative regret over 30 replications.

computational cost by updating the estimate only $O(\log T)$ times. The high-probability upper bound of the regret is tight and matches the lower bound up to logarithmic factors when the number of phases is small. Numerical studies demonstrate a good performance of the proposed method.

## Broader Impact

In this work, we present a novel algorithm for sequential decision. The main point is that the proposed method has low computational cost while achieving comparable performance to existing methods. The work mainly focuses on theoretical development of the algorithm, and uses simulated data for empirical evaluation. We believe that this work does not involve any ethical issue, and has no direct societal consequence.

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

# A Appendix

## A.1 Proofs

### A.1.1 Preliminaries

**Lemma A.1.1. (Lemma 12 of Abbasi-Yadkori et al., 2011)** *Let $X, Y$, and $Z$ be positive semi-definite matrices such that $Z = X + Y$. Then, we have,*

$$\sup_{x \neq 0} \frac{x^T Z x}{x^T Y x} \leq \frac{det(Z)}{det(Y)}.$$

**Lemma A.1.2. Elliptical Potential Lemma (Lemma 11 of Abbasi-Yadkori et al., 2011)** *Let $V_0 \in \mathbb{R}^{d \times d}$ be positive definite and $v_1, \cdots, v_n \in \mathbb{R}^d$ be a sequence of vectors with $||v_t||_2 \leq 1$ for all $1 \leq t \leq n$. Define $V_t = V_0 + \sum_{\tau=1}^{t} v_\tau v_\tau^T$. Then,*

$$\sum_{t=1}^{n} \left(1 \wedge v_t^T V_{t-1}^{-1} v_t\right) \leq 2\log\left(\frac{det(V_n)}{det(V_0)}\right).$$

### A.1.2 Proof of Theorem 4.1

**Theorem 4.1. Regret of AP-UCB**. *Suppose assumptions A1, A2, A3, and A4 hold. If we set $\alpha = 2R\sqrt{\log\left(\frac{2KT}{\delta}\right)} + \sqrt{\lambda}$ and $\beta = R\sqrt{3d\log\left(\frac{T}{\delta}\right)} + \sqrt{\lambda}$ for some $\delta \in (0, 1)$, we have with probability at least $1 - 2\delta$,*

$$R(T) \leq \sqrt{16dT\left\{2M_1 A R^2 \log\left(\frac{2KT}{\delta}\right) + 3M_2 E R^2 \log E \log\left(\frac{T}{\delta}\right)\right\} + 16T\lambda}$$
$$= O\left(\sqrt{dT(\log K + \log T)}(M_1 + M_2)\right).$$

*Proof.* We first have,

$$R(T) = \sum_{m=1}^{M} \sum_{t=t_{m-1}+1}^{t_m} regret(t) \leq \sqrt{T \sum_{m=1}^{M} \sum_{t=t_{m-1}+1}^{t_m} regret(t)^2}, \tag{4}$$

where the inequality follows from Jensen's inequality. Suppose $m$ is the $\bar{m}$-th independent phase. For any $t \in [t_{m-1}+1, t_m]$, the arm $a(t)$ is chosen based on $\hat{\mu}_{\bar{m}-1}$ and $B_{\bar{m}-1}$ from the $(\bar{m}-1)$-th independent phase. Thus for any $t \in [t_{m-1}+1, t_m]$, we have with probabiltiy at least $1 - \delta/t^2$,

$$b_{a^*(t)}(t)^T \mu \leq b_{a^*(t)}(t)^T \hat{\mu}_{\bar{m}-1} + \alpha \sqrt{b_{a^*(t)}(t)^T B_{\bar{m}-1}^{-1} b_{a^*(t)}(t)}$$
$$\leq b_{a(t)}(t)^T \hat{\mu}_{\bar{m}-1} + \alpha \sqrt{b_{a(t)}(t)^T B_{\bar{m}-1}^{-1} b_{a(t)}(t)}$$
$$\leq b_{a(t)}(t)^T \mu + 2\alpha \sqrt{b_{a(t)}(t)^T B_{\bar{m}-1}^{-1} b_{a(t)}(t)}, \tag{5}$$

where the first and third inequalities are due to Lemma 3.1 and the second inequality is due to the arm selection mechanism. Applying the union bound, we have with probability at least $1 - \sum_{t=t_{m-1}+1}^{t_m} \delta/t^2$,

$$
\begin{aligned}
\sum_{t=t_{m-1}+1}^{t_m} regret(t)^2 &\leq \sum_{t=t_{m-1}+1}^{t_m} 4\alpha^2 b_{a(t)}(t)^T B_{\bar{m}-1}^{-1} b_{a(t)}(t) \\
&= 4\alpha^2 trace\Big(B_{\bar{m}-1}^{-1} \sum_{t=t_{m-1}+1}^{t_m} b_{a(t)}(t) b_{a(t)}(t)^T\Big) \\
&\leq 4\alpha^2 trace\Big(B_{\bar{m}-1}^{-1} B_{\bar{m}}\Big),
\end{aligned}
\tag{6}
$$

where $B_{\bar{m}} = \lambda I_d + \sum_{t=t_{m-1}+1}^{t_m} b_{a(t)}(t) b_{a(t)}(t)^T$. Due to lines 5–9 in Algorithm 2, the matrix $B_{\bar{m}}$ without the last sample $b_{a(t_m)}(t_m)$ satisfies

$$
trace\Big\{B_{\bar{m}-1}^{-1}\big(B_{\bar{m}} - b_{a(t_m)}(t_m) b_{a(t_m)}(t_m)^T\big)\Big\} \leq Ad.
$$

Then,

$$
\begin{aligned}
trace(B_{\bar{m}-1}^{-1} B_{\bar{m}}) &\leq Ad + b_{a(t_m)}(t_m)^T B_{\bar{m}-1}^{-1} b_{a(t_m)}(t_m) \\
&\leq Ad + \frac{1}{\lambda},
\end{aligned}
\tag{7}
$$

due to Assumption A2 and the fact $B_{\bar{m}-1} \succcurlyeq \lambda I_d$.

On the other hand, when $m$ is an adaptive phase, arms are chosen based on $\tilde{\mu}_{m-1}$ and $B'_{m-1}$ constructed from all cumulative samples up to the $(m-1)$-th phase. Under similar arguments as in (5) but with Lemma 3.2 instead of Lemma 3.1, we have with probability at least $1 - \sum_{t=t_{m-1}+1}^{t_m} \delta/t^2$,

$$
\sum_{t=t_{m-1}+1}^{t_m} regret(t)^2 \leq \sum_{t=t_{m-1}+1}^{t_m} 4\beta^2 b_{\tilde{a}(t)}(t)^T B'^{-1}_{m-1} b_{\tilde{a}(t)}(t).
$$

Since we trivially have $regret(t) \leq 2$ and $\beta \geq 1$, we also have,

$$
\sum_{t=t_{m-1}+1}^{t_m} regret(t)^2 \leq \sum_{t=t_{m-1}+1}^{t_m} 4\beta^2 \big(1 \wedge b_{\tilde{a}(t)}(t)^T B'^{-1}_{m-1} b_{\tilde{a}(t)}(t)\big).
\tag{8}
$$

We follow the lines of Abbasi-Yadkori et al. (2011) to further bound (8). For any $t \in [t_{m-1}+1, t_m]$, let $B(t-1) = B'_{m-1} + \sum_{\tau=t_{m-1}+1}^{t-1} b_{\tilde{a}(\tau)}(\tau) b_{\tilde{a}(\tau)}(\tau)^T$. Then

$$
\begin{aligned}
(8) &\leq \sum_{t=t_{m-1}+1}^{t_m} 4\beta^2 \big(1 \wedge b_{\tilde{a}(t)}(t)^T B(t-1)^{-1} b_{\tilde{a}(t)}(t)\big) \frac{det(B(t-1))}{det(B'_{m-1})} \\
&\leq \sum_{t=t_{m-1}+1}^{t_m} 4\beta^2 \big(1 \wedge b_{\tilde{a}(t)}(t)^T B(t-1)^{-1} b_{\tilde{a}(t)}(t)\big) E \\
&\leq 8\beta^2 E \log\Big(\frac{det(B'_m)}{det(B'_{m-1})}\Big),
\end{aligned}
\tag{9}
$$

where the first inequality is due to Lemma A.1.1 and the fact that $B(t-1) \succcurlyeq B'_{m-1}$, the second inequality is due to line 21 of Algorithm 2, and the third inequality follows from Lemma A.1.2. Due to lines 21–25 in Algorithm 2, $B'_m$ without the last sample $b_{\tilde{a}(t_m)}(t_m)$ satisfies,

$$
det\Big(B'_m - b_{\tilde{a}(t_m)}(t_m) b_{\tilde{a}(t_m)}(t_m)^T\Big) \leq E det(B'_{m-1}).
$$

Then,

$$
\begin{aligned}
det(B'_m) &= det\Big(B'_m - b_{\tilde{a}(t_m)}(t_m)b_{\tilde{a}(t_m)}(t_m)^T\Big) \\
&\quad \times \Big\{1 + b_{\tilde{a}(t_m)}(t_m)^T\Big(B'_m - b_{\tilde{a}(t_m)}(t_m)b_{\tilde{a}(t_m)}(t_m)^T\Big)^{-1}b_{\tilde{a}(t_m)}(t_m)\Big\} \\
&\leq det\Big(B'_m - b_{\tilde{a}(t_m)}(t_m)b_{\tilde{a}(t_m)}(t_m)^T\Big)\Big(1 + \frac{1}{\lambda}\Big) \\
&\leq \Big(1 + \frac{1}{\lambda}\Big)Edet(B'_{m-1}).
\end{aligned}
\tag{10}
$$

Due to (4), (6), (7), (8), (9), and (10) and applying the union bound, we have with probability at least $1 - \sum_{m=1}^{M}\sum_{t=t_{m-1}+1}^{t_m}\delta/t^2$,

$$
R(T) \leq \sqrt{T\{M_1 4\alpha^2(Ad + 1/\lambda) + M_2 8\beta^2 E\log(E + E/\lambda)\}},
$$

where

$$
\sum_{m=1}^{M}\sum_{t=t_{m-1}+1}^{t_m}\frac{\delta}{t^2} = \delta + \sum_{t=2}^{T}\frac{\delta}{t^2} \leq \delta + \int_{t=1}^{\infty}\frac{\delta}{t^2}dt = 2\delta.
$$

Plugging in the definition of $\alpha$ and $\beta$ gives the theorem. $\qquad\square$

### A.1.3 Proof of Lemma 4.2

**Lemma 4.2.** *In the AP-UCB algorithm, we have*

$$
M_1 \leq \log_C\big(1 + T/d\lambda\big), \quad M_2 \leq d\log_E\big(1 + T/d\lambda\big).
$$

*Proof.* Derivation of the bound of $M_1$ is given in the main text. The upper bound of $M_2$ follows from the phase-switching condition for the adaptive phase given in line 21 of Algorithm 2. If $m$ is an adaptive phase, we have,

$$
det(B'_m) \geq Edet(B'_{m-1}).
$$

If $m$ is an independent phase, we have at least,

$$
det(B'_m) \geq det(B'_{m-1}),
$$

since $B'_m \succcurlyeq B'_{m-1}$. Therefore, $det(B'_M) \geq E^{M_2}det(\lambda I_d)$ with $det(B'_M) \leq \big(\lambda + \frac{T}{d}\big)^d$ and $det(\lambda I_d) = \lambda^d$. Thus,

$$
M_2 \leq d\log_E\big(1 + T/d\lambda\big).
$$

$\qquad\square$

### A.2 Experiment Details

We follow the design of Chu et al. (2011) to generate the contexts and the linear parameter $\mu$. We set $K = 2$ and $d = 11$. We divide the $T = 2000$ rounds into $h = (d-1)/2$ groups of $T' = T/h$ rounds such that each group has a different best arm. Time $t$ belongs to group $r \in [0, 1, \cdots, h-1]$ if the remainder of dividing $(t-1)$ by $h$ is $r$. When $t$ belongs to group $r$, we let $b_1(t)$ have 0.5 in the first and $(2r+2)$-th components, and 0 in the remaining components. On the other hand, we let $b_2(t)$ have 0.5 in the first and $(2r+3)$-th components, and 0 in the remaining components. The parameter $\mu$ has value 0.5 in the first component, $10/\sqrt{T'}$ in either the $(2r+2)$-th or $(2r+3)$-th coordinate for each group $r$, and 0 in the remaining components.

We generate the errors $\eta_i(t)$'s independently from the normal distribution $\mathcal{N}(0, 1)$. We run the experiments with $\beta \in [0.1, 0.6, 1.1, \cdots, 4.6]$ and $\alpha \in [0.1, 0.2, \cdots, 0.9]$, and report the results of the values that incur minimum median regret over 30 experiments.

