# OpenReview forum: "Adaptively Phased Algorithm for Linear Contextual Bandits"
_TMLR — Rejected by TMLR_

### Review · Reviewer_upwX · 2022-10-27

**Summary Of Contributions:**

This paper focuses on the linear contextual bandit problem with the finite-arm setting. The classical algorithm on linear contextual bandit usually needs prior knowledge of the number of round $T$. To deal with this limitation, this work proposes a novel phased algorithm called AP-UCB and obtains a regret bound $O(d^{\alpha}\sqrt{T \log T(\log K + \log T)})$. In addition, experiment results show that the AP-UCB algorithm outperforms other baselines, which supports the efficiency of the proposed algorithms.

**Broader Impact Concerns:**

There is no concern about the broader impact.

**Requested Changes:**

1. According to weakness 2, comparing the AP-UCB algorithm with the double trick on the previous algorithm is better.

2. According to weakness 3, it is necessary to make a clear definition for all assumptions (e.g., need prior knowledge for all future decision sets).

**Strengths And Weaknesses:**

Strengths:

1. This paper is well-written and easy to follow.

2. Experiments support the efficiency of the proposed algorithms.

Weakness:

1. As the author mentions, the key contribution of AP-UCB is removing the requirement of time horizon T. However, according to Theorem 4.1, in order to obtain a sub-linear regret guarantee, we need to set $\alpha, \beta =O(\sqrt{\log T})$. Thus, it seems that AP-UCB still needs prior knowledge about T.

2. The double trick can deal with the unknown horizon T, and it is unclear why we need to use a novel algorithm. Are there some advantages for AP-UCB, compared with the double trick on the previous algorithm?

3. For algorithm 2, if condition (C2) does not hold, we need to go back to the beginning of the current phase, and the previous chosen action will not be pushed. In this case, the algorithm needs to know all future actions at the beginning, which seems impossible with an un-fixed decision set. In addition, if the algorithm knows all future action sets, then the number of the different sets is just the time horizon $T$. Therefore, it is unreasonable to know future action sets without prior knowledge about the horizon $T$.

4. The regret guarantee for AP-UCB is $O(d^{\alpha}\sqrt{T \log T(\log K + \log T)})$, where $\alpha\in [1/2,1]$, and there is no guarantee for the $\alpha$. In the worst case, the regret is $O(d\sqrt{T \log T(\log K + \log T)})$, which matches the OFUL algorithm. In addition, according to the experiment result, most episodes are adaptive phases, and it seems that the regret will close to $O(d\sqrt{T \log T(\log K + \log T)})$.

---

> ### Author Response · Authors · 2022-12-09
> **Response**
>
> Response to weakness 1: As the reviewer pointed out, the proposed values of $\alpha$ and $\beta$ in Theorem 4.1 involve the unknown horizon $T$. However, we would like to clarify that $\alpha, \beta=O(\mathrm{log}T)$ is just a sufficient condition. The regret bound of the theorem still holds when we use time-varying values  $\alpha_t=2R\sqrt{\mathrm{log}\left(\frac{2Kt}{\delta}\right)}$ and $\beta_t=\left(R\sqrt{3d\mathrm{log}\left(\frac{t}{\delta}\right)}+\sqrt{\lambda}\right),$ where $T$ is replaced with current time step $t$. The proof follows in a straightforward manner from the inequalities in Lemmas 3.1 and 3.2 which are stated in terms of $\alpha_t$ and $\beta_t$ instead of $T$-dependent $\alpha$ and $\beta$. We will clarify on this point in the revision. While in SupLinUCB or SupLinRel algorithm, $T$ plays a crucial role in determining the number of phases, we do not use $T$ for any elements of our algorithm. Setting conservatively very high $T$ could result in wasting lots of samples.
>
>
> Response to weakness 2: The doubling trick can indeed be applied when $T$ is unknown, and the regret bound will have the same order as SupLinUCB. However, SupLinUCB is known to have a poor empirical performance even for the case where $T$ is known. Many studies have shown that the fully adaptive LinUCB outperforms SupLinUCB in practice even though LinUCB has $\sqrt{d}$-times higher worst-case regret bound. The doubling trick is known to waste many samples in practice, which may exacerbate the problem of poor performance. Experiments show that our algorithm performs well in practice as well.

---

> ### Author Response · Authors · 2022-12-09
> **Reponse (continued)**
>
> Response to weakness 3: We respectfully disagree with the claim that the algorithm needs to know all future actions at the beginning of each phase. The algorithm needs to see the contexts of all potential actions, not the actions. Knowing the contexts ahead does not mean that the algorithms knows the time horizon. Also, at the start of each phase, the algorithm needs to see the contexts only until the time step where the condition $trace(B_{\bar{m}-1}^{-1}B)\leq Ad$ is violated. This time step is much smaller than the total horizon $T$. Therefore, as long as the contexts are given by an oblivious adversary, the decision set needs not to be fixed.
>
> Response to weakness 4: Although we do not provide a theoretical guarantee that $\alpha$ is strictly smaller than 1 in this work, our methodology enables a novel decomposition of the regret which replaces the $O(d\sqrt{\mathrm{log}T})$ term in the bound  $O(d\sqrt{T\mathrm{log}T(\mathrm{log}K+\mathrm{log}T)})$ of RS-UCB with $O(\sqrt{d(M_1+M_2)})$, where $M_1$ and $M_2$ are the number of independent and adaptive phases respectively. Since $M_1\leq log_C(1+T/d\lambda)$ and $M_2\leq d~log_E(1+T/d\lambda)$, the worst case regret bound is $O(d\sqrt{T(\mathrm{log}K+\mathrm{log}T)\mathrm{log}(1+T/d)})$. However, $M_1$ and $M_2$ are inversely proportional to each other, i.e., $M_2$ decreases as $M_1$ increases and vice versa. Therefore, the worst case regret bound is achieved only when $M_2$ attains maximum and $M_1=0$. Thus, our method improves the regret bound of RS-UCB in terms of $d$ from $d$ to $d^{\alpha}$ where $\alpha=[1/2,1]$. The more independent phases, the smaller $\alpha$ becomes. We remind that the condition for the $m$-th phase to be an independent phase is to have $det(B_{m-1}^{-1}B)>C^d$ (condition (C2) in the text) at the moment when $trace(B_{m-1}^{-1}B)\leq Ad$ (condition (C1) in the text) is violated, i.e., as soon as $trace(B_{m-1}^{-1}B)\geq Ad$, where $B_{m-1}$ is the gram matrix of the $(m-1)$-th phase and $B$ is the gram matrix of the current phase. When we combine these two conditions (C1) and (C2),
> it becomes the  inequalities on the arithmetic mean and geometric mean of the eigenvalues of $B_{m-1}^{-1}B$. Using the relationship between the arithmetic mean and geometric mean of positive values(Tung, 1975), we can prove that $det(B_{m-1}^{-1}B)\geq \left(A-(1-\rho)\lambda_{\mathrm{max}}-\rho\lambda_{\mathrm{min}}+\lambda_{\mathrm{min}}^{\rho}\lambda_{\mathrm{max}}^{1-\rho}\right)^d$ as soon as $trace(B_{m-1}^{-1}B)\geq Ad$, where $\lambda_{\mathrm{min}}$ and $\lambda_{\mathrm{max}}$ are minimum and maximum eigenvalues of $B_{m-1}^{-1}B$ and $\rho=\mathrm{log}\left[\lambda_{\mathrm{max}}/(\lambda_{\mathrm{max}}-\lambda_{\mathrm{min}})\mathrm{log}(\lambda_{\mathrm{max}}/\lambda_{\mathrm{min}})\right]/\mathrm{log}(\lambda_{\mathrm{max}}/\lambda_{\mathrm{min}})$. Therefore, the condition for having an independent phase is satisfied when $A-(1-\rho)\lambda_{\mathrm{max}}-\rho\lambda_{\mathrm{min}}+\lambda_{\mathrm{min}}^{\rho}\lambda_{\mathrm{max}}^{1-\rho}\geq C$. A sufficient condition is known to be $A-\lambda_{\mathrm{max}}+\lambda_{\mathrm{min}}\geq C$. Since we can set the constant $C$ arbitrarily small, the probability that the condition is satisfied is strictly greater than 0. Therefore in many cases, $\alpha$ is less than 1.
> Experiments support this claim, where independent phases occur and result in reducing the regret bound as compared to RS-UCB.
>
> Our work is the first to utilize the trace along with the determinant in the stopping condition of phases. Using both the trace and determinant enables to introduce independent phases which contribute to reducing the regret bounds of linear bandit algorithms. While SupLinUCB and SupLinRel algorithms provide theoretical guarantee that $\alpha=\frac{1}{2}$, they mostly have much higher regret than LinUCB and RS-UCB in practice. We believe that our work contributes in resolving the existing trade-off between theoretical and empirical performance.
>
> References: Tung, S. H. (1975). On lower and upper bounds of the difference between the arithmetic and the geometric mean. Mathematics of Computation, 29(131), 834-836.

---

### Review · Reviewer_uK4W · 2022-10-31

**Summary Of Contributions:**

**Summary:**

This paper proposes a novel phased algorithm for linear contextual bandits, which does not require a priori knowledge of $T$, and instead constructs the phases in an adaptive way. The authors show that the proposed algorithm has a regret upper bound of order $O(d^{\alpha} \sqrt{T \log T (\log K + \log T)})$ where $1/2 \leq \alpha \leq 1$. The proposed algorithm can be regarded as a generalization of Rarely Switching OFUL [Abbasi-Yadkori et al., 2011] by applying a tight confidence bound in each phase obtained through the samples in the same phase.



**Broader Impact Concerns:**

I do not have any concern on the ethical implications of this work.

**Requested Changes:**

Please see the weaknesses mentioned above.

**Strengths And Weaknesses:**

**Strengths:**

1. This paper studies a classic and important problem in the bandit literature, i.e., linear contextual bandits. The authors design a phase-based algorithm, which does not require the prior knowledge of timestep $T$ and reduces the computation costs. A rigorous regret analysis is provided. This paper is well executed.

**Weaknesses:**

1. Linear contextual bandit is a well-studied problem in the bandit literature. As mentioned by the related work section, there have been several works which design phase-based algorithms with near-optimal regret bounds for this problem. [Li et al. 2019] refines the SupLinUCB and achieves a bound of $O(\sqrt{d T \log T \log K}(\log \log T )^{\gamma})$ with $\gamma>0$. It is unclear to me how large the improvement the bound $O(d^{\alpha} \sqrt{T \log T (\log K + \log T)})$ ($1/2 \leq \alpha \leq 1$) in this paper achieves over that in [Li et al. 2019]. I think it will be more clear if the authors can present a table to give a detailed comparison of settings, algorithm features and regret bounds with existing linear bandit works.

2. It is unclear to me how to choose the parameters $B$ and $C$ of Algorithm AP-UCB to satisfy the condition (C2) and achieve a good regret bound as stated in Lemma 4.2.
In addition, I think the results of Theorem 4.1 and Lemma 4.2 lack a clear discussion and comparison. Specifically, the values of $M_1$ and $M_2$ in Theorem 4.1 and the comparison to [Li et al. 2019] are not clear. While the authors give Lemma 4.2 and briefly mention that they can avoid the extra $O(\sqrt{d})$ factor by letting $M_2$ small and $M_1$ large, the authors do not clearly discuss how to achieve this in Algorithm AP-UCB and compare this result to existing results, e.g., [Li et al. 2019]. It is difficult for me to understand how large improvement this paper achieves.

3. It would be better if the authors can display the conditions (C1), (C2) by formal equations, instead of just mentioning them by in-line sentences, because these two conditions are important for the discussion on the theoretical performance of Algorithm AP-UCB.

**Overall Review:**

Overall, while I think that the algorithm design and regret analysis in this paper are well-executed, the idea of adaptive phases is not novel and has been similarly used in existing linear bandit works. The theoretical results in this paper lack a clear discussion and comparison with existing linear bandit works. It is difficult for me to understand how large the improvement this paper achieves on regret performance. Due to these reasons, I give weak rejection.

---

> ### Author Response · Authors · 2022-12-09
> **Response**
>
> Response to weakness 1: The underlying assumptions of the algorithms are all the same for SupLinUCB, SupLinRel, VCL-SupLinUCB(Li et al., 2019), LinUCB, RS-UCB, and our algorithm(AP-UCB). Our improvements are two-fold, (i) in regret bounds and (ii) in algorithmic features.
>
> As for the regret bound, our main contribution is to replace the $O(d\sqrt{\mathrm{log}T})$ term in the regret bound of LinUCB and RS-UCB with $O(\sqrt{d(M_1+M_2)})$, where $M_1$ and $M_2$ are inversely proportional to each other. Our bound is  $O(\sqrt{dT(\mathrm{log}K+\mathrm{log}T)\mathrm{log}(1+T/d)})$ when $M_2$ is a constant multiple of $M_1$, which matches the bound of VCL-SupLinUCB(Li et al., 2019) up to logarithmic terms. Please see our response to Weakness 2 for detailed comparison of regret bounds.
>
> As for algorithmic features, we drop the requirement of the knowledge of the time horizion $T$. In SupLinUCB, SupLinRel, and VCL-SupLinUCB the knowledge of $T$ plays a crucial role in determining the number of phases. When $T$ is unknown, setting conservatively very high $T$ results in unnecessarily many phases which makes waste lots of samples. SupLinUCB, SupLinRel and VCL-SupLinUCB are known to have bad empirical performance due to wasting of samples even with correct specification of $T$. Our algorithm does not use $T$ in any step as it determines the number of phases adaptively. Despite less requirements, our algorithm outperforms existing algorithms in practice as well, as shown in our synthetic experiments.

---

> ### Author Response · Authors · 2022-12-09
> **Response (continued)**
>
> Response to weakness 2: In condition (C2), $B$ refers to the gram matrix of the $m$-th phase. We assume that the reviewer referred to the constants $A$ and $C$ in conditions (C1) and (C2) respectively. We only require the constants $A$ and $C$ to be independent of $d$, $T$ and $K$. The same goes for the constant $E$ used in line 21 of the algorithm. In this case, we always have $M1\leq \mathrm{log}_C(1+T/d\lambda)$ and $M_2\leq d\mathrm{log}_E(1+T/d\lambda)$ as stated in Lemma 4.2. Combining the lemma with Theorem 4.1, we obtain a worst case regret bound of order $O(d\sqrt{T(\mathrm{log}K+\mathrm{log}T)\mathrm{log}(1+T/d)})$ which matches the regret bound of LinUCB and RS-UCB.
>
> Although our regret bound is $\sqrt{d\mathrm{log}(T/d)}$ times higher than the bound of VCL-SupLinUCB(Li et al. 2019) in the worst case, we emphasize that this bound is attained when $M_2$ is at maximum. Our main contribution is to introduce independent phases to the naive RS-UCB which consists of adaptive phases only. This introduction enables to replace the $O(d\sqrt{\mathrm{log}T})$ term in the regret bound of RS-UCB with $O(\sqrt{d(M_1+M_2)})$, where $M_1$ and $M_2$ are inversely proportional to each other, i.e., $M_2$ decreases as $M_1$ increases and vice versa. Our bound matches the bound of RS-UCB only when $M_2$ is at maximum. On the other hand, when $M_1$ is at maximum, $M_2=0$ and the regret bound is  $O(\sqrt{dT(\mathrm{log}K+\mathrm{log}T)\mathrm{log}(1+T/d)})$ which matches the bound of VCL-SupLinUCB up to logarithmic terms. Although we do not provide specific conditions for $A, C, E$ to have maximum $M_1$ and minimum $M_2$ in this paper, we empirically demonstrate a special case where the regret of our algorithm is lower than LinUCB and RS-UCB. This improvement is due to the reduction (by nearly half) in the number of adaptive phases as compared to RS-UCB. We remind that the condition for the $m$-th phase to be an independent phase is to have $det(B_{m-1}^{-1}B)>C^d$ (condition (C2) in the text) at the moment when $trace(B_{m-1}^{-1}B)\leq Ad$ (condition (C1) in the text) is violated, i.e., as soon as $trace(B_{m-1}^{-1}B)\geq Ad$, where $B_{m-1}$ is the gram matrix of the $(m-1)$-th phase and $B$ is the gram matrix of the current phase. When we combine these two conditions (C1) and (C2),
> it becomes the  inequalities on the arithmetic mean and geometric mean of the eigenvalues of $B_{m-1}^{-1}B$. Using the relationship between the arithmetic mean and geometric mean of positive values(Tung, 1975), we can prove that $det(B_{m-1}^{-1}B)\geq \left(A-(1-\rho)\lambda_{\mathrm{max}}-\rho\lambda_{\mathrm{min}}+\lambda_{\mathrm{min}}^{\rho}\lambda_{\mathrm{max}}^{1-\rho}\right)^d$ as soon as $trace(B_{m-1}^{-1}B)\geq Ad$, where $\lambda_{\mathrm{min}}$ and $\lambda_{\mathrm{max}}$ are minimum and maximum eigenvalues of $B_{m-1}^{-1}B$ and $\rho=\mathrm{log}\left[\lambda_{\mathrm{max}}/(\lambda_{\mathrm{max}}-\lambda_{\mathrm{min}})\mathrm{log}(\lambda_{\mathrm{max}}/\lambda_{\mathrm{min}})\right]/\mathrm{log}(\lambda_{\mathrm{max}}/\lambda_{\mathrm{min}})$. Therefore, the condition for having an independent phase is satisfied when $A-(1-\rho)\lambda_{\mathrm{max}}-\rho\lambda_{\mathrm{min}}+\lambda_{\mathrm{min}}^{\rho}\lambda_{\mathrm{max}}^{1-\rho}\geq C$. A sufficient condition is known to be $A-\lambda_{\mathrm{max}}+\lambda_{\mathrm{min}}\geq C$. Since we can set the constant $C$ arbitrarily small, the probability that the condition is satisfied is greater than 0.
>
> SupLinUCB, SupLinRel and VCL-SupLinUCB are known to have very bad empirical performance despite their sharp regret bounds. On the other hand, LinUCB and RS-UCB have shown good performance in real applications. In this paper, we provide an algorithm which is both practical and has a regret bound that matches that of the practical LinUCB and RS-UCB only in the worst case.
>
> Response to weakness 3: Thank you for your suggestion. We definitely agree with you. We will state the conditions using formal equations in the revised paper.
>
> References: Tung, S. H. (1975). On lower and upper bounds of the difference between the arithmetic and the geometric mean. Mathematics of Computation, 29(131), 834-836.

---

> > ### Comment · Reviewer_uK4W · 2022-12-29
> > **Thank the authors for their responses**
> >
> > Thank you for your reply. I read the comments of other reviewers.
> >
> > While this paper contributes an improvement to the linear bandit problem, I do not think this improvement is significant, given the following facts: (i) existing linear bandit algorithms can already achieve an optimal regret up to a logarithmic factor, (ii) existing doubling tricks can drop the requirement of known $T$, and (iii) the improvement proposed by this paper is not clearly and strictly compared (it is not easy to quantify how large the improvement is compared to existing results).

---

### Review · Reviewer_Qebd · 2022-11-29

**Summary Of Contributions:**

This paper proposed a novel algorithm for linear contextual bandit, which does not require the time horizon $T$ as a prior knowledge. The upper bound of AP-UCB nearly matches the established lower bound by existing work. It also run experiments to evaluate the performance of the proposed AP-UCB algorithm.

**Broader Impact Concerns:**

I think there is no ethical concern.

**Requested Changes:**

As stated above, I have two suggestions for changes:
1. I appreciate that the instruction section describes some real-life motivation of the setting, i.e, a real-life scenario where we do not know the horizon $T$ and can only apply the AP-UCB algorithm.
1. Regarding (B1) and (B2) in Section 3, instead of “arms are chosen independently/adaptively….”, according to my understanding, it should be “rewards are generated independently/adaptively…”.
The authors may carefully check if this cause any analytical difference if I am correct.

**Strengths And Weaknesses:**

Strengths:
1. The paper is well organized and easy to read. The contribution is described clearly.
1. Although the time horizon is not an unexpected prior knowledge, it is still an advantage as the AP-UCB does not require such input.

Weaknesses:
1. I appreciate that the instruction section describes some real-life motivation of the setting, i.e, a real-life scenario where we do not know the horizon $T$ and can only apply the AP-UCB algorithm.
1. Regarding (B1) and (B2) in Section 3, instead of “arms are chosen independently/adaptively….”, according to my understanding, it should be “rewards are generated independently/adaptively…”.
The authors may carefully check if this cause any analytical difference if I am correct.

---

> ### Author Response · Authors · 2022-12-19
> **Response**
>
> Response to weakness 1: Thank you very much for your suggestion. One scenario with unknown time horizon can happen in personalized recommendation systems for shopping sites or movie sites. Whenever a new user comes in, a bandit algorithm  adapting to the specific user is applied for each user. While some users visit the site frequently for a long time, others might drop out early. Thus, the time horizon for each user is unknown. When the time horizon is unknown, either underestimation or overestimation of the time horizon can reduce the cumulative reward. We will describe this scenario in the Introduction section of the revised paper.
>
> Response to weakness 2: Thank you for your suggestion. We referred ``arms are chosen independently/adaptively" as follows.  Let us assume that we are in the $(m+1)$-th phase at the $(t+1)$-th round.
>  When we say 'independent', we use the estimates based on data from the $m$-th phase. When we say `adaptive', we use the estimates based on data from the first phase to $m$-th phase AND data from round 1 to $t$ of the $(m+1)$-th phase. Since in the independent case we do not use data from  round 1 to $t$ of the $(m+1)$-th phase, the arm selection indicator is independent of the rewards from round 1 to $t$ of the $(m+1)$-th phase. The same independence cannot be obtained in the adaptive case since the arm selection depends on the rewards from round 1 to $t$ of the $(m+1)$-th phase. In the independent case, the regression coefficient estimate $\hat{\mu}$ enjoys a tight prediction error bound as in Lemma 3.1. On the other hand, in the adaptive case, only a looser prediction error bound (Lemma 3.2) can be guaranteed.

---

### Decision · Action_Editors · 2023-01-07

**Recommendation:** Reject

**Comment:**

Reviewers in general find the paper easy to read, and do not have major technical questions. The main weakness lies in the unclear level of interests in the paper's findings. Reviewers are not convinced that the regret bound represents a meaningful improvement over knowing results (Li et al 2019 in particular). The a priori knowledge of the time horizon is more of a theoretical concern, and is already dealt with by the standard doubling trick. Considering these, reviewers find the contributions to have limited interests among the TMLR audience.

**Audience:**

Unclear audience, given unclear improvements over known results.

**Claims And Evidence:**

The paper provides an algorithm with a regret bound, without a priori knowledge of the time horizon. A complete proof is provided for the regret bound. The algorithm is also tested in simple experiments. Overall the evidence is clear to support the claims.